# LoRa-Over: A Matrix Decomposition-Based Over-Parameterization for Efficient LLM Fine-Tuning

## Abstract

The rise of large language models (LLMs) has revolutionized machine learning, yielding state-of-the-art results across various tasks through extensive pre-training. While full-parameter fine-tuning is the standard for adapting LLMs, its high storage and computational costs are major drawbacks. Parameter-efficient methods, such as LoRA, mitigate these issues by updating a small subset of parameters, but often sacrifice generalization performance. In this work, we introduce LoRa-Over (Over-Parameterization for Low-Rank Adaptation), a novel approach that enhances generalization by strategically over-parameterizing low-rank matrices during fine-tuning. Using matrix decomposition, LoRa-Over achieves near-lossless reconstruction and maintains inference efficiency. It employs static and dynamic strategies to selectively over-parameterize critical matrices, balancing computational cost and performance. LoRa-Over is validated on tasks such as natural language understanding (GLUE with T5-Base), dialogue generation (MT-Bench), mathematical reasoning (GSM8K), and code generation (HumanEval) using Llama 2-7B and Llama 3.1-8B models. Results show significant performance improvements over vanilla LoRA, demonstrating its potential as a scalable, efficient fine-tuning framework for diverse downstream applications. All the experimental codes will be released after the review period.

## 1 Introduction

The advent of modern large language models (LLMs) has fundamentally revolutionized the field of machine learning, achieving exceptional performance across a wide range of tasks by leveraging massive pre-training on trillion-token corpora (Devlin et al., 2019; Liu et al., 2019; Radford et al., 2019; Brown et al., 2020). A dominant strategy for adapting pre-trained LLMs to downstream applications is full-parameter fine-tuning (commonly referred to as full fine-tuning) (Qiu et al., 2020; Raffel et al., 2020), wherein all parameters of the model are updated using gradient-based optimization. While this approach delivers outstanding results, it presents significant challenges, particularly in terms of substantial storage demands and high computational costs, making it difficult to deploy in practice.

To overcome these limitations, parameter-efficient fine-tuning (PEFT) methods have emerged as a promising solution. PEFT techniques, such as Low-Rank Adaptation (LoRA) (Hu et al., 2022), aim to efficiently adapt large language models to specific domains by freezing the majority of pre-trained parameters and updating only a small subset. This approach alleviates the computational complexity and memory overhead associated with full fine-tuning while maintaining reasonable performance. However, the drastic reduction in trainable parameters can compromise the model's ability to generalize, often resulting in suboptimal fine-tuning outcomes compared to full fine-tuning.

In order to address this performance gap, we propose a novel approach that strategically over-parameterizes low-rank parameter matrices during fine-tuning, thereby enhancing the model's generalization capability. By leveraging matrix decomposition techniques (Henry & Hofrichter, 1992; Tucker, 1966; Oseledets, 2011), any given matrix can be decomposed into a product of smaller matrices. Singular Value Decomposition (SVD), for instance, is a widely used method in this domain. Matrix decomposition not only facilitates temporary parameter expansion during fine-tuning

but also allows for recombination of the decomposition components to restore the original model architecture after fine-tuning, thus preserving inference efficiency. While over-parameterizing low-rank matrices shows promise, it introduces two key challenges. First, it is essential to minimize information loss caused by matrix decomposition, as minor inaccuracies can accumulate and propagate through the stacked Transformer layers of LLMs, potentially degrading performance. Second, LLMs contain numerous low-rank parameter matrices, many of which are not equally critical across different tasks (Zhang et al., 2022; Voita et al., 2019). Over-parameterizing all such matrices indiscriminately would be both computationally expensive and inefficient. Therefore, it is crucial to identify and selectively over-parameterize the most important matrices.

To address these challenges, we adopt the Matrix Product Operator (MPO) (Pirvu et al., 2010) technique from quantum many-body physics as our primary matrix decomposition method. MPO can effectively factorize matrices of arbitrary dimensions into multi-scale tensor structures while ensuring near-lossless reconstruction through tensor recomposition (Gao et al., 2020). These properties make MPO particularly well-suited for the over-parameterization of low-rank matrices during fine-tuning. Using MPO, we design both static and dynamic strategies to adaptively over-parameterize significant low-rank matrices. Leveraging MPO, we implement adaptive over-parameterization through both static and dynamic strategies for selecting significant low-rank parameter matrices. For the static strategy, we estimate the importance of the low-rank parameter matrix based on the change in loss values when it is removed from a fine-tuned model (Voita et al., 2019). The top-$N$ matrices are then over-parameterized. The dynamic strategy computes the variation of gradients within several fine-tuning steps. This serves as an approximation to the aforementioned loss variation (Hou et al., 2020) method and dynamically guides the matrix over-parameterization process during fine-tuning. Building on these techniques, we introduce **LoRa-Over** (Over-Parameterization for Low-Rank Adaptation), a novel framework designed to improve the fine-tuning performance of LLMs. LoRa-Over utilizes MPO decomposition to over-parameterize low-rank matrices, enhancing the fine-tuning procedure while maintaining task-agnostic characteristics. This makes the framework adaptable to diverse downstream tasks.

We conduct extensive experiments to evaluate the efficacy of LoRa-Over. For natural language understanding tasks, we assess the performance of T5-Base Raffel et al. (2020) on the GLUE subset (Wang et al., 2018). For dialogue generation tasks, mathematical reasoning tasks and code generation tasks, we apply our method to Llama 2-7B (Touvron et al., 2023) and Llama 3.1-8B (Grattafiori et al., 2024), evaluating their performance on the MT-Bench dataset (Zheng et al., 2023), GSM8K dataset (Cobbe et al., 2021) and HumanEval dataset (Chen et al., 2021), respectively. Experimental results demonstrate that LoRa-Over significantly outperforms vanilla LoRA, achieving substantial performance gains. For instance, LoRa-Over consistently outperforms vanilla LoRA by 4.98% on the GLUE subset with T5-Base, and by 0.32, 11.07%, 5.00% on MT-Bench, GSM8K, and HumanEval with Llama 2-7B, respectively. Additionally, LoRa-Over also consistently outperforms vanilla LoRA by 0.11, 5.76%, 2.40% on MT-Bench, GSM8K, and HumanEval with Llama 3.1-8B, respectively. These results underscore the versatility and effectiveness of LoRa-Over in addressing the limitations of vanilla LoRA while preserving computational efficiency.

## 2 RELATED WORK

**Over-parameterization in Learning Process.** Over-parameterization has demonstrated its effectiveness in multiple dimensions of deep learning. Previous studies highlighted its utility in improving model initialization (Arpit & Bengio, 2019), optimizing training dynamics through improved convergence (Allen-Zhu et al., 2019b; Gao et al., 2021; Du et al., 2018), and enhancing generalization capabilities (Allen-Zhu et al., 2019a). Sparked by the hypothesis of lottery theory (Frankle & Carbin, 2018), subsequent research had further emphasized its potential to increase training efficiency Malach et al. (2020); Pensia et al. (2020) and improved model performance Chen et al. (2020); Brix et al. (2020); Prasanna et al. (2020). In particular, in-time over-parameterization strategies (Liu et al., 2021b) were employed to bridge the performance gap between sparse and dense network training regimes. Building on these foundations, we proposed to leverage over-parameterization as a principled approach to unlock the latent potential of the LoRA method, boosting their fine-tuning performance.

**Tensor Decomposition in Neural Network.** Tensor decomposition had emerged as a cornerstone technique for improving the efficiency of neural network training and inference. Its versatility is evident in applications ranging from model compression (Gao et al., 2020), lightweight fine-tuning (Liu et al., 2021a; Gao et al., 2023), and knowledge distillation (Zhan et al., 2024). Pioneering works had extensively utilized these methods to decompose parameter matrices, achieving significant compression ratios for linear layers (Novikov et al., 2015) and convolutional kernels (Garipov et al., 2016). Beyond model compression, recent advances further showcased their adaptability: MPO-based decomposition efficiently scales the MoE framework for dynamic model capacity (Gao et al., 2022), while parameterized tensor formats enable resource-conscious fine-tuning of ALBERT (Liu et al., 2021a). Diverging from conventional paradigms that prioritize dimensionality reduction, our work exploited tensor decomposition from an inverse perspective: we deployed it as a latent space bridge to implicitly over-parameterize adapters by mapping low-dimensional parameters to high-dimensional latent spaces during fine-tuning.

**Variants of LoRA in LLMs.** LoRA is a widely used technique for fine-tuning LLMs, significantly reducing their resource requirements. Given its efficiency, numerous variants had been developed to improve the original method, employing diverse approaches to enhance adaptability, stability, or task-specific performance. PiSSA (Meng et al., 2024) is a PEFT method for LLMs. It initialized a low-rank adapter using the principal singular components derived from the pre-trained weight matrix via SVD, contrasting with random initialization methods like LoRA. And rsLoRA (Kalajdzievski, 2023) performed the adaptation by dynamically adjusting the rank and scaling coefficients of low-rank matrices. This dynamic adaptation mechanism optimized adjustments based on real-time data distribution and model requirements during fine-tuning, ensuring stability in parameter updates to prevent both overfitting and underfitting. And AdaLoRA (Zhang et al., 2023) improved the allocation of trainable parameters in LoRA by dynamically assigning a tunable parameter budget based on the importance of each parameter. It parameterized incremental updates using SVD. This SVD-based parametrization avoided large-scale SVD computations while enabling efficient pruning of non-essential singular values in updates, thereby reducing resource overhead during adaptation. Furthermore, LoHA (Hyeon-Woo et al., 2021) and LoKr (Edalati et al., 2025) employed Hamiltonian and Kronecker products, respectively. Our approach expanded the parameter space by increasing the number of trainable parameters, incorporating a lossless MPO decomposition method to augment the parameter set.

## 3 PRELIMINARY

**Tensor** We denote a tensor $\mathcal{T}_{i_1, i_2, \ldots, i_m}$ as an array with $m$ indices, where $\{i_1, i_2, \ldots, i_m\}$ denotes the dimensions of the $m$ indices, respectively. The intrinsic tensor-based nature of both data and trainable parameters in deep learning establishes tensor representations as fundamental building blocks in neural networks.

**Tensor Product** As a fundamental construct in linear algebra, the tensor product formalism serves as a cornerstone in quantum mechanical analysis and also remains indispensable for both conceptual advances and computational protocols in many-body physics. Considering $\{\psi_i\}_{i=1}^p$ and $\{\phi_j\}_{j=1}^p$ are the orthonormal basis of tensors $\mathcal{T}^{(1)}$ and $\mathcal{T}^{(2)}$, respectively. The $\otimes$ denotes the tensor productFormally, the tensor contraction of $\mathcal{T}^{(1)} = \sum_{i=1}^p \alpha_i \psi_{i_1}$ and $\mathcal{T}^{(2)} = \sum_{j=1}^q \beta_j \phi_{i_2}$ is defined as follow:

$$\mathcal{T}^{(1)} \otimes \mathcal{T}^{(2)} = \sum_{i=1}^p \sum_{j=1}^q \alpha_i \beta_j \psi_{i_1} \otimes \phi_{i_2}. \tag{1}$$

**Tensor Decomposition** Tensor decomposition can be regarded as the inverse operation of the tensor product. The SVD algorithm serves as a widely utilized mathematical framework for tensor decomposition. Given a tensor $\mathcal{T} \in \mathbb{R}^{i_1 \times \cdots \times i_n}$, through an iterative sequence of $n$ SVD operations, the original tensor can be factorized into $n$ hierarchically structured local tensors $\{\mathcal{T}^{(k)}\}_{k=1}^n$. In contrast, the decomposed tensors can reconstruct the original tensor $\mathcal{T}$ by sequentially utilizing the tensor product operator.

# 4 METHOD

In this section, we first give a review of vanilla LoRA. Subsequently, we propose our LoRa-Over, which employs MPO to increase the number of low-rank matrix parameters and introduces the over-parameterized low-rank parameter matrices selection strategies.

## 4.1 REVISIT THE LoRA METHOD

LoRA (Hu et al., 2022) is a parameter-efficient fine-tuning framework that introduces low-rank matrices $A \in \mathbb{R}^{r \times d_2}$ and $B \in \mathbb{R}^{d_1 \times r} (r \ll min(d_1, d_2))$. During training, the pre-trained weight matrix $W_0$ remains frozen, while the model is updated by training the low-rank matrices $A$ and $B$, which extremely diminishes the number of trainable parameters, thus decreasing the computational cost of fine-tuning. In particular, the mathematical expression of LoRA is defined as

$$W = W^{(0)} + \Delta W = W^{(0)} + \frac{\alpha}{r} BA, \tag{2}$$

where $\alpha$ and $r$ are hyperparameters of scaling-factor and LoRA rank. Vanilla LoRA initializes $A$ with Kaiming normal distribution (He et al., 2015), while $B$ adopts zero initialization. This initialization strategy ensures $BA = 0$ at the beginning of training. Additionally, the low-rank matrices can be merged into the pre-trained matrix, and LoRA does not introduce any extra latency during the inference compared with full funetuning.

## 4.2 OVER-PARAMTERIZATION LOW RANK MATRICES VIA MPO DECOMPOSITION

To enhance the vanilla LoRA method by leveraging the benefits of over-parameterization during the fine-tuning process, the proposed approach utilizes the MPO, a tensor network decomposition technique, to expand the parameter space of the model. Specifically, the methodology first details the fundamental principles of MPO decomposition. Subsequently, we describe the adaptation and application of this MPO framework to construct over-parameterized low-rank parameter matrices. The overview of our approach is presented in Figure 1.

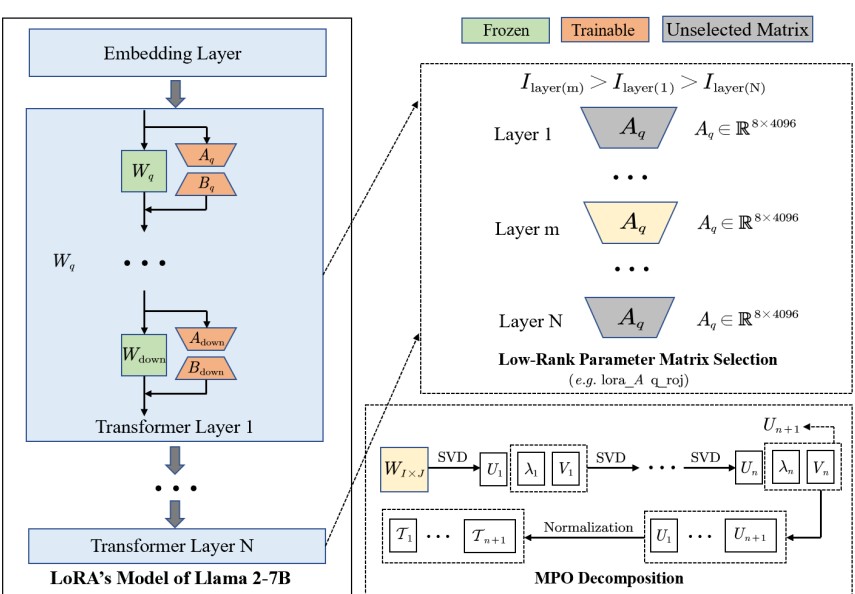

Figure 1: The overview of over-parameterization in LoRA fine-tuning LLMs. $I$ is defined as the estimated significance score for the parameter matrices. We also demonstrate a scenario wherein the low-rank parameter matrix $\mathbf{W}$ is selected for over-parameterization and is consequently decomposed into a set of higher-order tensors $\{\mathcal{T}^{(k)}\}_{k=1}^{n+1}$.

**Matrix Product Operator Decomposition.** MPO decomposition efficiently factorizes a parameter matrix $\mathbf{W} \in \mathbb{R}^{I \times J}$ into a sequential product of multiple tensors (Gao et al., 2020). Formally, the MPO decomposition of a parameter matrix $\mathbf{W} \in \mathbb{R}^{I \times J}$ yields an ordered sequence of $m$ local tensors $\{\mathcal{T}^{(k)}\}_{k=1}^{m}$ can be denoted as

$$\text{MPO}(\mathbf{W}) = \bigotimes_{k=1}^{m} \mathcal{T}^{(k)}[d_{k-1}, i_k, j_k, d_k]. \tag{3}$$

Where the tensor $\mathcal{T}^{(k)}[d_{k-1}, i_k, j_k, d_k]$ is a 4th-order tensor with size $[d_{k-1}, i_k, j_k, d_k]$, in which $\prod_{k=1}^{m} i_k = I, \prod_{k=1}^{m} j_k = J$, and $d_0 = d_m = 1$. The concept of a bond, introduced to link two sequence tensors, has been adopted following the work of (Pirvu et al., 2010). The dimension of the bond $d_k$ is denoted as

$$d_k = \min(\prod_{p=1}^{k} i_p \times j_p, \prod_{p=k+1}^{n} i_p \times j_p). \tag{4}$$

A deterministic mapping process from the parameter matrix $\mathbf{W}$ to multiple high-order tensors $\{\mathcal{T}^{(k)}\}_{k=1}^{m}$ is defined by the given tensor sizes $\{d_k\}_{k=1}^{m}$, $\{i_k\}_{k=1}^{m}$, and $\{j_k\}_{k=1}^{m}$. Through iterative matrix reshaping and SVD decomposition (Henry & Hofrichter, 1992) executed on $m$-turns, the MPO process continuously reduces the size of the parameter matrix and sequentially generates decomposed tensors. Reshaping is applied during the $k$-th turn to the matrix of the previous turn $\mathbf{W}_{k-1}$, transforming it into the matrix $\mathbf{W}'_{k-1}$ with the first dimension $d_{k-1} \times i_k \times j_k$. Subsequently, we decompose it via SVD as

$$\mathbf{U}\lambda\mathbf{V}^{\text{T}} = \text{SVD}(\mathbf{W}'_{k-1}) \tag{5}$$

where $\mathbf{U}$ and $\mathbf{V}$ are complex unitary matrices, $\lambda$ is a rectangular diagonal matrix with non-negative real numbers on the diagonal. Inspired by truncated SVD methods (Henry & Hofrichter, 1992), corresponding to the $d_k$ largest singular values, the first $d_k$ columns of $\mathbf{U}$ form the decomposed tensor $\mathcal{T}^{(k)}$, which is subsequently reshaped to match the dimensions of $[d_{k-1}, i_k, j_k, d_k]$. Furthermore, $\lambda\mathbf{V}^{\text{T}}$ is assigned as the output parameter matrix $\mathbf{W}_k$ for the decomposition in the subsequent turns. Following $m$-turn iterations, the decomposition results in a set of multiple high-order tensors $\{\mathcal{T}^{(k)}\}_{k=1}^{m}$. The original parameter matrix $\mathbf{W}$ can be recovered almost losslessly by contracting these tensors sequentially (Gao et al., 2020). You can find the comprehensive algorithm in Algorithm S.1 of Appendix B.

**Over-parameterizing Low Rank Matrices.** Utilizing the MPO methodology, we strategically amplify low rank matrices parameterization scales during fine-tuning to exploit advantages inherent in structured over-parameterization. More precisely, the MPO method enables the decomposition of selected low-rank parameter matrices into multiple tensors according to Eq. equation 3. The values of $\{d_k\}_{k=1}^{m}$, $\{i_k\}_{k=1}^{m}$, and $\{j_k\}_{k=1}^{m}$ govern the increase in parameter number in matrix W after MPO decomposition. The detailed added parameter number $N_{add}$ derives from the following calculation procedure.

$$N_{add} = \sum_{k=1}^{m} i_k j_k d_{k-1} d_k - \prod_{k=1}^{m} i_k j_k. \tag{6}$$

Following the formalism of Eq. equation 4, the determination of $\{d_k\}_{k=1}^{m}$ is based on $\{i_k; j_k\}_{k=1}^{m}$. Control over the number of added parameters is achieved by adjusting the values of $\{i_k; j_k\}_{k=1}^{m}$ within the MPO decomposition strategy. Consequently, the fine-tuning process allows the adoption of MPO on selected low-rank parameter matrices to generate the corresponding multiple tensors. This methodology enables scaling of the model's total parameter number, effectively enhancing its over-parameterization. After achieving convergence by fine-tuning the over-parameterized low-rank parameter matrices, tensor contraction is performed on the decomposed tensors to reconstruct the model's parameter matrices. The resulting model preserves identical parameter number and inference latency to the original, while retaining over-parameterization benefits during fine-tuning.

## 4.3 OVER-PARAMETERIZED LOW RANK MATRICES SELECTION

The MPO decomposition method is recognized for its computational tractability and representational flexibility. However, its application for over-parameterizing all low-rank parameter matrices remains

computationally expensive. To concentrate the benefits of over-parameterization on the most critical parameters, the approach selectively applies MPO decomposition only to the most important low-rank parameter matrices. Consequently, a dual strategy is proposed: a static selection method, which preidentifies significant low-rank parameter matrices prior to fine-tuning, and a dynamic selection method, which continuously identifies and selects critical low-rank parameter matrices throughout the fine-tuning process.

**Static Selection Strategy.** The proposed static selection strategy entails the pre-computation of importance scores for all candidate low-rank parameter matrices prior to the fine-tuning phase. Subsequently, this strategy employs the MPO formalism to over-parameterize exclusively the top-$N$ low-rank parameter matrices. Consequently, the architecture of the resulting over-parameterized LoRA model remains fixed throughout the subsequent fine-tuning process. Inspired by network pruning methods (Molchanov et al., 2016; Voita et al., 2019), we quantify importance scores for individual low-rank parameter matrices by measuring the resulting perturbation in training loss $\mathcal{L}_{\mathbf{W}}$ after surgical removal of each low-rank parameter matrix $\mathbf{W}$. This metric is theoretically grounded in the principle that parameters exerting significant influence on predictive accuracy will inherently manifest elevated loss differentials when excised, as such low-rank parameter matrices fundamentally support the correct assignment of labels (Voita et al., 2019). Thus, the importance score $I_{\mathbf{W}}$ of a low-rank parameter matrix $\mathbf{W}$ can be calculated as

$$I_{\mathbf{W}} = |\mathcal{L}_{\mathbf{W}} - \mathcal{L}_{\mathbf{W}=\mathbf{0}}|, \tag{7}$$

where $\mathcal{L}_{\mathbf{W}=\mathbf{0}}$ represents the loss value after zeroing $\mathbf{W}$. To compute the loss, fine-tuning must commence from identical pre-trained parameters as our baseline prior to low-rank parameter matrix integration. Crucially, low-rank parameter matrices originating from distinct modules inherently vary in size and functionality, rendering direct performance comparisons invalid. To address this methodological challenge, we first classify all low-rank parameter matrices into module-specific categories, where each group corresponds to a single modular component spanning $L$ layers. Within every such group, the top-$N$ performing low-rank parameter matrices are subsequently isolated for over-parameterization.

**Dynamic Selection Strategy.** We propose a dynamic selection strategy that continuously computes importance scores to identify significant low-rank parameter matrices for instantaneous over-parameterization during fine-tuning. This approach dynamically assesses the importance of change with respect to the optimization of the whole model. The approximation of the importance score can be obtained by performing the first-order Taylor expansion on Eq. equation 7

$$I_{\mathbf{W}} = \left| \mathcal{L}_{\mathbf{W}} - \left( \mathcal{L}_{\mathbf{W}} - \frac{\partial \mathcal{L}}{\partial \mathbf{W}}(\mathbf{W} - \mathbf{0}) + R_{\mathbf{W}=\mathbf{0}} \right) \right| \approx \left| \frac{\partial \mathcal{L}}{\partial \mathbf{W}} \mathbf{W} \right|, \tag{8}$$

The omission of part $R_{\mathbf{W}=\mathbf{0}}$ allows the computation of the important score through the absolute gradients of the parameter matrix. Throughout the fine-tuning process, accumulation occurs for the absolute gradients across all low-rank parameter matrices. Dynamically computed using the Eq. equation 8, importance scores trigger over-parameterization of top-$N$ low rank parameter matrices in categorized groups at $t$-steps. The iterative cycle of this process persists until the selection of N low-rank parameter matrices is achieved per group. And we also present a detailed algorithm for our selection strategy. (see Algorithm S.2 in Appendix B)

## 5 EXPERIMENTS

In this section, we assess the performance of LoRa-Over on various benchmark datasets. Initially, we employ comprehensive experiments on the General Language Understanding Evaluation (GLUE) benchmark (Wang et al., 2018) with the T5-Base (Raffel et al., 2020) model. Subsequently, we evaluate dialogue, arithmetic reasoning, and coding abilities using the Llama 2-7B (Touvron et al., 2023) model and the Llama 3.1-8B (Grattafiori et al., 2024) model. We then report the results and provide a thorough analysis.

**Baseline Methods.** We compare LoRa-Over with several baselines to demonstrate its effectiveness. *Full fine-tuning* is the most prevalent adaptation approach, which updates all model

parameters but requires substantial computational resources. $LoRA$ (Hu et al., 2022) constitutes a parameter-efficient fine-tuning method that introduces a low-rank parameter matrix product $BA$ into linear layers, where $A$ is initialized by Kaiming initialization and $B$ is initialized to zero. $rsLoRA$ (Kalajdzievski, 2023) incorporates a novel scaling factor to enhance the stability of the LoRA scale. $PiSSA$ (Meng et al., 2024) employs SVD (Henry & Hofrichter, 1992) in the weight matrix $W$ at initialization, and uses larger singular values for better performance. $LoRA+$ (Hayou et al., 2024) applies different learning rates for the low-rank matrix $A$ and $B$ in LoRA. $OLoRA$ (Büyükakyüz, 2024) incorporates orthonormal initialization for the adaptation matrices. $DoRA$ (Liu et al., 2024) enhances expressiveness by adding learnable magnitudes. $AdaLoRA$ (Zhang et al., 2023) dynamically prunes nonessential weights via SVD, reallocating rank to diminish GPU memory. Furthermore, we conduct a comparative analysis between our method and SVD, a classical matrix factorization technique, which is viable for over-parameterizing our model. Concretely, this methodological substitution replaces MPO with SVD within our method, implementing comprehensive over-parameterization across all the low-rank parameter matrices during fine-tuning. The implementation details can be found in the appendix.

## 5.1 Experiments on Natural Language Understanding

**Models and Datasets.** To present a comprehensive overview of the performance of our proposed LoRa-Over, we fine-tune the T5-Base model on a subset of GLUE datasets, including MNLI, SST-2, CoLA, QNLI, and MRPC. Performance is evaluated on the corresponding validation sets using accuracy as the metric.

| Method | MNLI 393k | SST-2 67k | CoLA 8.5k | QNLI 105k | MRPC 3.7k | Avg. |
|---|---|---|---|---|---|---|
| Full | 86.33 | 94.75 | 80.70 | 93.19 | 84.56 | 87.91 |
| LoRA | 85.30 | 94.04 | 69.35 | 92.96 | 68.38 | 82.08 |
| *LoRA Variants with Original Structure* | | | | | | |
| PiSSA | 85.75 | 94.07 | 74.27 | 93.15 | 76.31 | 84.71 |
| rsLoRA | 85.73 | 94.19 | 72.32 | 93.12 | 52.86 | 79.64 |
| LoRA+ | 85.81 | 93.85 | 77.53 | 93.14 | 74.43 | 84.95 |
| *LoRA Variants with Modified Structure* | | | | | | |
| DoRA | 85.67 | 94.04 | 72.04 | 93.04 | 68.08 | 82.57 |
| AdaLoRA | 85.45 | 93.69 | 69.19 | 91.66 | 68.14 | 81.62 |
| LoRa-Over-SVD | 85.33 | 94.27 | 72.39 | 93.03 | 68.40 | 82.68 |
| LoRa-Over-MPO | 85.42 | 94.38 | 79.19 | 93.19 | 77.70 | 85.98 |
| LoRa-Over-MPO$_S$ | 85.59 | 94.50 | 78.43 | 93.23 | 74.51 | 85.25 |
| LoRa-Over-MPO$_D$ | **85.84** | **94.61** | **79.29** | **93.45** | **82.11** | **87.06** |

Table 1: Performance comparison using T5-Base on the GLUE benchmark (in percent). **Bold** scores represents the best performance, underline scores indicates the second-best performance. For all the results, we report the mean values of five runs using different random seeds.

**Results.** As shown in Table 1, LoRa-Over-MPO$_D$ achieves superior performance on all datasets. It achieves the highest accuracy on MNLI (85.84), SST-2 (94.61), CoLA (79.29), QNLI (93.45), and MRPC (82.11). LoRa-Over-MPO$_D$'s average score (87.06) surpasses all other methods and surpasses vanilla LoRA with a margin of 4.98, demonstrating outstanding adaptability and generalization. These results show the effectiveness of our approach. Notably, for small datasets, CoLA and MRPC, our method shows highly strong performance, highlighting that it utilizes small training data effectively.

## 5.2 Experiments on Natural Language Generation

**Models and Datasets.** We evaluate the performance of LoRa-Over on Llama 2-7B and Llama 3.1-8B. For dialogue generation, we train our model on a 52k subset of the WizardLM dataset (Xu et al., 2024) and evaluate it using the MT-Bench dataset. The quality of the model responses is assessed using GPT-4, with the first-turn score reported as the evaluation metric. For mathematical reasoning, we train our model on a 100k subset of MetaMathQA (Yu et al., 2023). The model's performance

is evaluated on the GSM8K test set, with accuracy reported as the metric. For code generation, we train our model on a 100k subset of the CodeFeedback dataset (Zheng et al., 2024) and evaluate it on the HumanEval dataset, and the model performance is quantified via the PASS@1 metric.

| Method | MTBench | GSM8K | HumanEval | Avg |
|---|---|---|---|---|
| **Llama 2-7B (Touvron et al., 2023)** | | | | |
| Full | 5.30 | 59.36 | 35.31 | 33.32 |
| LoRA | 5.61 | 42.08 | 14.76 | 20.82 |
| PiSSA | 5.30 | 44.54 | 16.02 | 21.95 |
| rsLoRA | 5.25 | 45.62 | 16.01 | 22.29 |
| OLoRA | 5.30 | 43.29 | 17.22 | 21.94 |
| LoRA+ | 5.71 | 52.11 | 18.17 | 25.33 |
| AdaLoRA | 5.57 | 50.72 | 17.80 | 24.70 |
| DoRA | **5.97** | 53.07 | 19.75 | 26.26 |
| LoRa-Over-SVD | 5.23 | 44.81 | 15.00 | 21.68 |
| LoRa-Over-MPO | 5.66 | 49.51 | 17.32 | 24.16 |
| LoRa-Over-MPO$_S$ | 5.63 | 47.76 | 17.07 | 23.49 |
| LoRa-Over-MPO$_D$ | 5.92 | **53.15** | **19.76** | **26.28** |
| **Llama 3.1-8B (Grattafiori et al., 2024)** | | | | |
| Full | 5.88 | 73.69 | 51.63 | 43.73 |
| LoRA | 6.15 | 67.78 | 43.09 | 39.01 |
| PiSSA | 6.08 | 68.56 | 44.10 | 39.58 |
| rsLoRA | 6.18 | 68.36 | **45.78** | 40.11 |
| OLoRA | 6.13 | 68.54 | 43.29 | 39.32 |
| LoRA+ | **6.35** | 71.29 | 44.51 | 40.72 |
| AdaLoRA | 6.19 | 70.63 | 41.46 | 39.43 |
| DoRA | 6.24 | 69.17 | 43.70 | 39.70 |
| LoRa-Over-SVD | 6.01 | 68.92 | 42.68 | 39.20 |
| LoRa-Over-MPO | 6.21 | 71.42 | 43.41 | 40.35 |
| LoRa-Over-MPO$_S$ | 6.16 | 71.19 | 43.17 | 40.17 |
| LoRa-Over-MPO$_D$ | 6.26 | **73.54** | 45.49 | **41.76** |

Table 2: Performance comparison using Llama 2-7B and Llama 3.1-8B on MT-Bench, GSM8K, and HumanEval (in percent). **Bold** and underline indicate the highest and second-highest scores, respectively. For all the results, we report the mean values of five runs using different random seeds.

**Results.** The results presented in Table 2 reveal a consistent performance advantage of LoRa-Over over other baseline methods in most tasks. For Llama 2-7B, LoRa-Over-MPO$_D$ demonstrates exceptional performance on both the GSM8K and HumanEval datasets. Although slightly underperforming DoRA on MT-Bench by just 0.05 points, LoRa-Over-MPO$_D$'s average score (26.28) surpasses all baselines. Specifically, LoRa-Over-MPO$_D$ achieves the highest score on GSM8K (53.15) and HumanEval (19.76), substantially surpassing vanilla LoRA, indicating superior performance in mathematical reasoning and code generation. For Llama 3.1-8B, LoRa-Over-MPO$_D$ exhibits superior performance compared to the baseline methods. It achieves the highest score on GSM8K (73.54), achieving performance comparable to full finetuning (73.69). On MT-Bench, LoRa-Over-MPO$_D$ scores 6.26, placing second to LoRA+. On HumanEval, it ranks second with 45.49, just behind rsLoRA. LoRa-Over-MPO$_D$'s average performance is 41.76, surpassing LoRA+'s 40.72 by 1.04 points. The results indicate that SVD typically demonstrates inferior performance compared to MPO among the two matrix decomposition methods. We observe that the dynamic strategy generally exhibits superior performance over the static strategy at identical parameter scales. The obtained results demonstrate that LoRa-Over-MPO$_D$ effectively improves training stability and delivers consistently robust performance in a variety of natural language generation tasks.

## 5.3 FURTHER ANALYSIS

Following this, a more detailed analysis is undertaken to rigorously examine the proposed approach.

**Hyper-parameters Tuning.** The performance of our method with a dynamic strategy is primarily influenced by two hyper-parameters: the total count of selected parameter matrices $N$ and their group count $n$, making their tuning crucial. A higher value of $N$ corresponds to a higher number of parameter matrices being selected and over-parameterized. Conversely, a smaller $n$ indicates that a larger subset of matrices is over-parameterized simultaneously in a single operation. To investigate the effects of these hyperparameters, we perform an empirical analysis on the CoLA and MRPC datasets using the T5-base model. As illustrated in Figure 2, model performance improves steadily with increasing values of $N$ before eventually saturating. This trend suggests that an insufficient degree of over-parameterization fails to adequately capture the full representation capacity of the low-rank matrix. Furthermore, an excessively low value of $n$ is observed to adversely affect performance. This degradation may be attributed to the fact that an insufficient group size causes an excessive number of matrices to be over-parameterized simultaneously, which effectively reduces the dynamic strategy to a static approach.

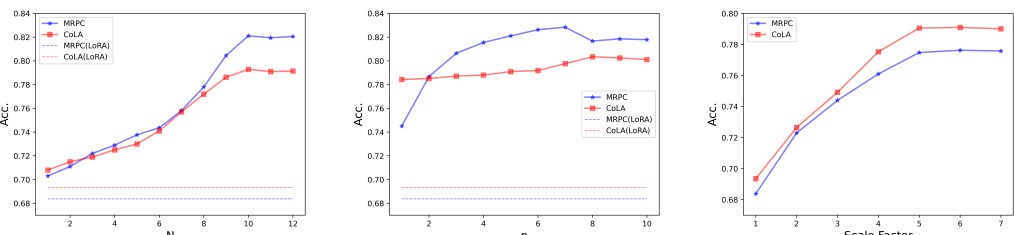

Figure 2: (**Left**)Performance comparison with varying total parameter matrices selection number $N$ and group number $n$ on MRPC and CoLA using T5-Base. (**Right**) Performance comparison under different parameter expansion scales on MRPC and CoLA tasks.

**Robustness Analysis.** Our approach leverages matrix decomposition to over-parameterize low-rank matrices but is prone to numerical instability, where small perturbations during decomposition can accumulate and cause significant errors. To address this, we employ MPO decomposition for near-lossless factorization, enhancing stability and reducing sensitivity to hyperparameter changes. We assess robustness by varying the learning rate on CoLA and MRPC tasks with a T5-Base model. Performance across learning rates 1.6e-4, 1.8e-4, 2e-4, 2.2e-4, 2.4e-4 (Table S.5 in the appendix) shows our method is resilient to changes, with 2e-4 achieving strong results, avoiding extensive hyperparameter tuning.

**Parameter Increasing Rate Analysis.** To enhance the over-parameterization of low-rank matrices, our method deliberately increases the number of trainable parameters during fine-tuning. Given that the proposed method provides a general and flexible framework for scaling trainable parameters, we systematically evaluate its performance across multiple parameter counts. Based on LoRA's model of T5-Base, we systematically increase the trainable parameters via over-parameterization (from $1\times$ to $7\times$) and assess the performance on the MRPC and CoLA tasks. Figure 2 demonstrates a consistent, monotonic improvement in model performance as the parameter scale increases. Empirical results indicate that performance improvement asymptotically approaches a plateau at the $7\times$ parameter scale. A potential interpretation is that this scale exhausts the primary benefits of over-parameterization for this specific architecture and task set.

## 6 CONCLUSION

This paper introduces a novel over-parameterization framework designed to increase the parameter number of low-rank matrices, specifically during fine-tuning to leverage the benefits of increased model capacity. Our method incorporates the MPO approach to decompose low-rank parameter matrices into high-order tensors to augment the number of parameters, and also employs static and dynamic strategies to select matrices for over-parameterization based on importance. Through extensive experiments, we have demonstrated that our method can boost the performance of LoRA fine-tuning significantly and narrow the gap between vanilla LoRA and full fine-tuning. Future work will explore enhanced tensor decomposition methods to optimize the over-parameterization of low-rank parameter matrices.

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
