# APPENDIX

## A DETAILS OF TENSORS

### A.1 TENSOR AND MATRIX PRODUCT OPERATORS

As introduced in Gao et al. (2020), a tensor is precisely characterized as follows:

**Tensor.** Let $D_1, D_2, ..., D_P \in N$ denote index upper bounds. A tensor $\mathcal{T} \in \mathbb{R}^{D_1, D_2, ..., D_P}$ of order $P$ is a $P$-way array where elements $\mathcal{T}[d_1, d_2, ..., d_P]$ are indexed by $d_p \in \{1, 2, ..., D_P\}$ for $1 \le p \le P$.

**Matrix Product Operator.** The concept of bond dimension $d_k$ is defined as follows:

$$d_k = \min(\prod_{p=1}^{k} i_p \times j_p, \prod_{p=k+1}^{n} i_p \times j_p). \tag{S.1}$$

We can observe that is will be large in the middle and small on both sides by equation S.1. A detailed algorithm for MPO decomposition can be found in Algorithm hdak of Appendix hdakj. The MPO representation decomposes $M$ into a product of $n$ local tensors:

$$M_{i_1 i_n, j_1 j_n} = \mathcal{T}^{(1)}[i_1, j_1]\mathcal{T}^{(n)}[i_n, j_n] \tag{S.2}$$

where $\mathcal{T}^{(p)}[i_p, j_p]$ is a $D_{p-1} \times D_p$ matrix with $D_p$ the virtual basis dimension on the bond linking $\mathcal{T}^{(p)}$ and $\mathcal{T}^{(p+1)}$ with $D_0 = D_n = 1$.

### A.2 THEOREM

**Theorem 1.** *Suppose that the tensor $\mathbf{W}^{(k)}$ of matrix W that is satisfy*

$$\mathbf{W} = \mathbf{W}^{(k)} + \mathbf{E}^{(k)}, D(\mathbf{W}^{(k)}) = d_k, where \quad \|\mathbf{E}^{(k)}\|_F^2 = \epsilon_k^2, k = 1, ..., d-1. \tag{S.3}$$

*Then MPO(**W**) with the $k$-th bond dimension $d_k$ upper bound of truncation error satisfy:*

$$\|\mathbf{W} - MPO(\mathbf{W})\|_F \le \sqrt{\sum_{k=1}^{d-1} \epsilon_k^2} \tag{S.4}$$

*Proof.* The proof is by induction. For $n = 2$ the statement follows from the properties of the SVD. Consider an arbitrary $n > 2$. Then the first unfolding $\mathbf{W}^{(1)}$ is decomposed as:

$$\mathbf{W}^{(1)} = \mathbf{U}_1 \lambda_1 \mathbf{V}_1 + \mathbf{E}^{(1)} = \mathbf{U}_1 \mathbf{B}^{(1)} + \mathbf{E}^{(1)} \tag{S.5}$$

where $\mathbf{U}_1$ is of size $r_1 \times i_1 \times j_1$ and $\|\mathbf{E}^{(1)}\|_F^2 = \epsilon_1^2$. The matrix $\mathbf{B}_1$ is naturally associated with a $(n-1)$-dimensional tensor $\mathcal{B}^{(1)}$ with elements $\mathcal{B}^{(1)}(\alpha, i_2, j_2, ..., i_n, j_n)$, which will be decomposed further. This means that $\mathbf{B}_1$ will be approximated by some other matrix $\hat{\mathbf{B}}_1$. From the properties of the SVD it follows that $\mathbf{U}_1^T \mathbf{E}^{(1)} = 0$, and thus

$$\|\mathbf{W} - \mathcal{B}^{(1)}\|_F^2$$
$$= \|\mathbf{W}_1 - \mathbf{U}_1\hat{\mathbf{B}}_1\|_F^2$$
$$= \|\mathbf{W}_1 - \mathbf{U}_1(\hat{\mathbf{B}}_1 + \mathbf{B}_1 - \mathbf{B}_1)\|_F^2$$
$$= \|\mathbf{W}_1 - \mathbf{U}_1\mathbf{B}_1\|_F^2 + \|\mathbf{U}_1(\hat{\mathbf{B}}_1 - \mathbf{B}_1)\|_F^2 \tag{S.6}$$

and since $\mathbf{U}_1$ has orthonormal columns,

$$\|\mathbf{W} - \mathcal{B}^{(1)}\|_F^2 \le \epsilon_1^2 + \|\mathbf{B}_1 - \hat{\mathbf{B}}_1\|_F^2. \tag{S.7}$$

and thus it is not difficult to see from the orthonormality of columns of $\mathbf{U}_1$ that the distance of the $k$-th unfolding ($k = 2, ..., d_k - 1$) of the $(d-1)$-dimensional tensor $\mathcal{B}^{(1)}$ to the $d_k$-th rank matrix cannot be larger than $\epsilon_k$. Proceeding by induction, we have

$$\|\mathbf{B}_1 - \hat{\mathbf{B}}_1\|_F^2 \le \sum_{k=2}^{d-1} \epsilon_K^2, \tag{S.8}$$

combine with Eq. equation S.7, this complets the proof.

# B  ALGORITHMS

The MPO pseudocode is shown in Algorithm S.1.

---

**Algorithm S.1** MPO decomposition for a matrix.

---

**Input:** matrix $\mathbf{M}$, the number of local tensors $m$.
**Output** : MPO tensor list $\{\mathcal{T}_{(s)}\}_{s=1}^{m}$.
 1: **for** $s = 1 \to m$ **do**
 2:     $\mathbf{M}[I, J] \to \mathbf{M}[d_{s-1} \times i_s \times j_s, -1]$
 3:     $\mathbf{U}\lambda\mathbf{V}^{\mathrm{T}} = \mathrm{SVD}(\mathbf{M})$
 4:     $\mathbf{U}[d_{s-1} \times i_s \times j_s, d_s] \to \mathcal{U}[d_{s-1}, i_s, j_s, d_s]$
 5:     $\mathcal{T}^{(s)} := \mathcal{U}$
 6:     $\mathbf{M} := \lambda\mathbf{V}^{\mathrm{T}}$
 7: **end for**
 8: $\mathcal{T}^{(s)} := \mathbf{M}$
 9: Normalization
10: **return** $\{\mathcal{T}_{(k)}\}_{k=1}^{n}$

---

The over-parameterized matrices selection pseudocode is shown in Algorithm S.2

---

**Algorithm S.2** Fine-tuning a model with our OPF.

---

**Input:** Low rank parameter matrices set of a model $\{\mathbf{W}\}$.
 1: Divide $\{\mathbf{W}\}$ into several groups by module.
 2: **if** is Static Strategy **then**
 3:     LoRA fine-tuning the model until converged.
 4:     Compute $I_{\mathbf{W}}$ for $\{\mathbf{W}\}$ using equation 7.
 5:     Sort $\{\mathbf{W}\}$ in each group according to $I_{\mathbf{W}}$.
 6:     Perform MPO on the top-$N$ matrices.
 7:     Train the other PLM until converged.
 8: **else**
 9:     Define $S = \{\}$
10:     **while** Len($S$)$< N$ **do**
11:         Train the model for $t$ steps.
12:         Compute $I_{\mathbf{W}}$ for $\{\mathbf{W}\}$ using equation 8.
13:         Sort $\{\mathbf{W}\}$ in each group according to $I_{\mathbf{W}}$.
14:         Add top-$n$ matrices into $S$, and perform MPO.
15:     **end while**
16:     Continually train the model until converged.
17: **end if**

---

# C  ADDITIONAL EXPERIMENT DETAILS

In this paper, we propose the MPO decomposition as a method to increase the parameters of the model. Using Eq. 3, an MPO can be specified as:

$$\mathcal{T}_{i_1,i_2,\ldots,i_n}^{j_1,j_2,\ldots,j_n}(D) \tag{S.9}$$

We pre-compute the significance scores of all parameter matrices before fine-tuning and subsequently over-parameterize the top-$N$ ones using the MPO technique. The significant score can be calculated by Eq. equation 7 and Eq. equation 8.

## C.1  IMPLEMENTATION DETAILS

To ensure a fair comparison, we maintain the experimental setup of GoRA (He et al., 2025) and adopt baseline performances reported by them. By default, we fine-tune the converged models using the AdamW optimizer (Loshchilov & Hutter, 2017) with $\beta_1 = 0.9$, $\beta_2 = 0.999$, $\epsilon = 1e - 8$. We implement a cosine learning rate schedule with a warmup ratio of 0 and set the rank $r = 8$ and $\alpha = 16$. For natural language understanding tasks, we fine-tune T5-base (Raffel et al., 2020) with a

sequence length of 128, a training batch size of 32, and a weight decay of 0. For natural language generation tasks, we fine-tune Llama 2-7B (Touvron et al., 2023) with a sequence length of 1024, a training batch size of 32, a weight decay of 0, and fine-tune Llama 3.1-8B (Grattafiori et al., 2024) with a sequence length of 512, a training batch size of 64, a weight decay of 5e-4. All generation is performed with $top\_p = 0.95$ and temperature $T = 0.8$. All experiments use single-epoch training. For T5-Base and Llama 2-7B, LoRA target is all linear modules except embedding layer, layer norm and language model head. For Llama 3.1-8B, we train attention modules linear components. More details are presented in Appendix C. In our work, we adopt two types of GPUs: NVIDIA A100 GPUs and NVIDIA H100 GPUs. For the Natural Language Understanding tasks, all computations are performed on the A100 GPUs. For the Natural Language Generation tasks, all computations are executed on the H100 GPUs.

## C.2 EXPERIMENTS ON NATURAL LANGUAGE UNDERSTANDING

| Dataset | LR | MPO_LR | split number | top-$N$ | eval step |
|---|---|---|---|---|---|
| **LoRa-Over-SVD** | | | | | |
| **MNLI** | 2e-4 | Null | Null | Null | 100 |
| **QNLI** | 2e-4 | Null | Null | Null | 80 |
| **CoLA** | 2e-4 | Null | Null | Null | 5 |
| **SST-2** | 2e-4 | Null | Null | Null | 50 |
| **MRPC** | 2e-4 | Null | Null | Null | 5 |
| **LoRa-Over-MPO** | | | | | |
| **MNLI** | 2e-4 | Null | Null | Null | 100 |
| **QNLI** | 2e-4 | Null | Null | Null | 80 |
| **CoLA** | 2e-4 | Null | Null | Null | 5 |
| **SST-2** | 2e-4 | Null | Null | Null | 50 |
| **MRPC** | 2e-4 | Null | Null | Null | 5 |
| **LoRa-Over-MPO$_S$** | | | | | |
| **MNLI** | 2e-4 | 1.7e-4 | Null | 10 | 100 |
| **QNLI** | 2e-4 | 1.7e-4 | Null | 10 | 80 |
| **CoLA** | 2e-4 | 1.7e-4 | Null | 10 | 5 |
| **SST-2** | 2e-4 | 1.7e-4 | Null | 10 | 50 |
| **MRPC** | 2e-4 | 1.7e-4 | Null | 10 | 5 |
| **LoRa-Over-MPO$_D$** | | | | | |
| **MNLI** | 2e-4 | 1.7e-4 | 5 | 10 | 100 |
| **QNLI** | 2e-4 | 1.7e-4 | 5 | 10 | 80 |
| **CoLA** | 2e-4 | 1.7e-4 | 5 | 10 | 5 |
| **SST-2** | 2e-4 | 1.7e-4 | 5 | 10 | 50 |
| **MRPC** | 2e-4 | 1.7e-4 | 5 | 10 | 5 |

Table S.1: Hyperparameter setup of LoRa-Over for GLUE benchmark (T5-Base). "LR" denotes the learning rate. "Null" denotes the parameter is useless.

| Matrix Shape | LoRa-Over-SVD | LoRa-Over-MPO |
|---|---|---|
| (768,8) | $\mathcal{T}_{2,4}^{24,32}(D)$ | $\mathcal{T}_{2,1,1,1,1,1,1,1,1,1,1,1,1,1,1,4}^{24,1,1,1,1,1,1,1,1,1,1,1,1,1,1,32}(D)$ |
| (8,768) | $\mathcal{T}_{24,32}^{2,4}(D)$ | $\mathcal{T}_{24,1,1,1,1,1,1,1,1,1,1,1,1,1,1,32}^{2,1,1,1,1,1,1,1,1,1,1,1,1,1,1,4}(D)$ |
| (3072,8) | $\mathcal{T}_{2,4}^{48,64}(D)$ | $\mathcal{T}_{2,1,1,1,1,1,1,1,1,1,1,1,1,1,1,4}^{48,1,1,1,1,1,1,1,1,1,1,1,1,1,1,64}(D)$ |
| (8,3072) | $\mathcal{T}_{48,64}^{2,4}(D)$ | $\mathcal{T}_{48,1,1,1,1,1,1,1,1,1,1,1,1,1,1,64}^{2,1,1,1,1,1,1,1,1,1,1,1,1,1,1,4}(D)$ |

Table S.2: Summary of the MPO structure (T5-Base).

The hyperparameters of LoRa-Over using the T5-Base model and the MPO structure of the T5-Base model are presented in the Table S.1 and the Table S.2, respectively.

| Datasets | LR | MPO_LR | split number | top-$N$ | eval step |
|---|---|---|---|---|---|
| **Llama 2-7B (Touvron et al., 2023)** | | | | | |
| **LoRa-Over-SVD** | | | | | |
| **MT-Bench** | 1.9e-5 | Null | Null | Null | 100 |
| **GSM8K** | 1.9e-5 | Null | Null | Null | 100 |
| **HumanEval** | 1.9e-5 | Null | Null | Null | 100 |
| **LoRa-Over-MPO** | | | | | |
| **MT-Bench** | 6e-5 | Null | Null | Null | 100 |
| **GSM8K** | 6e-5 | Null | Null | Null | 100 |
| **HumanEval** | 6e-5 | Null | Null | Null | 100 |
| **LoRa-Over-MPO$_S$** | | | | | |
| **MT-Bench** | 6.2e-5 | 5.6e-5 | Null | 28 | 100 |
| **GSM8K** | 6.8e-5 | 6.1e-5 | Null | 28 | 100 |
| **HumanEval** | 6e-5 | 5.3e-5 | Null | 28 | 100 |
| **LoRa-Over-MPO$_D$** | | | | | |
| **MT-Bench** | 6.2e-5 | 5.6e-5 | 5 | 28 | 100 |
| **GSM8K** | 6.8e-5 | 6.1e-5 | 7 | 28 | 100 |
| **HumanEval** | 6e-5 | 5.3e-5 | 5 | 28 | 100 |
| **Llama 3.1-8B (Grattafiori et al., 2024)** | | | | | |
| **LoRa-Over-SVD** | | | | | |
| **MT-Bench** | 6e-5 | Null | Null | Null | 50 |
| **GSM8K** | 6e-5 | Null | Null | Null | 50 |
| **HumanEval** | 6e-5 | Null | Null | Null | 50 |
| **LoRa-Over-MPO** | | | | | |
| **MT-Bench** | 6e-5 | Null | Null | Null | 50 |
| **GSM8K** | 6e-5 | Null | Null | Null | 50 |
| **HumanEval** | 1e-4 | Null | Null | Null | 50 |
| **LoRa-Over-MPO$_S$** | | | | | |
| **MT-Bench** | 1e-4 | 9.4e-5 | Null | 30 | 50 |
| **GSM8K** | 1e-4 | 9.5e-5 | Null | 31 | 50 |
| **HumanEval** | 1e-4 | 7e-5 | Null | 28 | 50 |
| **LoRa-Over-MPO$_D$** | | | | | |
| **MT-Bench** | 1e-4 | 9.4e-5 | 5 | 30 | 50 |
| **GSM8K** | 1e-4 | 9.5e-5 | 5 | 31 | 50 |
| **HumanEval** | 1e-4 | 7e-5 | 3 | 28 | 50 |

Table S.3: Hyperparameter setup of LoRa-Over for Llama 2-7B and Llama 3.1-8B model. "LR" denote the learning rate. "Null" denote the parameter is useless.

### C.3 EXPERIMENTS ON NATURAL LANGUAGE GENERATION

The hyperparameters of LoRa-Over using Llama 2-7B and Llama 3.1-8B models are presented in the Table S.3. The MPO structure of Llama 2-7B and Llama 3.1-8B model is presented in the Table S.4.

### C.4 COMPARISON OF DIFFERENT LEARNING RATES

Our approach leverages matrix decomposition to over-parameterize low-rank matrices but faces numerical instability, as small perturbations during decomposition can accumulate and compromise model integrity. To address this, we employ MPO decomposition, which ensures near-lossless factorization, enhancing stability and reducing sensitivity to hyperparameter variations. We evaluate robustness by testing learning rates 1.6e-4, 1.8e-4, 2e-4, 2.2e-4, 2.4e-4 on CoLA and MRPC tasks with a T5-Base model (Table S.5). Results show our method is resilient to learning rate changes, with 2e-4 achieving strong performance, minimizing the need for extensive hyperparameter tuning.

| Matrix Shape | MT-Bench | GSM8K | HumanEval |
|---|---|---|---|
| **Llama 2-7B (Touvron et al., 2023)** | | | |
| **LoRa-Over-SVD** | | | |
| (4096,8) | $\mathcal{T}_{2,4}^{64,64}(D)$ | $\mathcal{T}_{2,4}^{64,64}(D)$ | $\mathcal{T}_{2,4}^{64,64}(D)$ |
| (8,4096) | $\mathcal{T}_{64,64}^{2,4}(D)$ | $\mathcal{T}_{64,64}^{2,4}(D)$ | $\mathcal{T}_{64,64}^{2,4}(D)$ |
| (11008,8) | $\mathcal{T}_{2,4}^{86,128}(D)$ | $\mathcal{T}_{2,4}^{86,128}(D)$ | $\mathcal{T}_{2,4}^{86,128}(D)$ |
| (8,11008) | $\mathcal{T}_{86,128}^{2,4}(D)$ | $\mathcal{T}_{86,128}^{2,4}(D)$ | $\mathcal{T}_{86,128}^{2,4}(D)$ |
| **LoRa-Over-MPO** | | | |
| (4096,8) | $\mathcal{T}_{2,1,1,1,1,1,1,4}^{64,1,1,1,1,1,1,64}(D)$ | $\mathcal{T}_{2,1,1,1,1,1,1,4}^{64,1,1,1,1,1,1,64}(D)$ | $\mathcal{T}_{2,1,1,1,1,1,1,4}^{64,1,1,1,1,1,1,64}(D)$ |
| (8,4096) | $\mathcal{T}_{64,1,1,1,1,1,1,64}^{2,1,1,1,1,1,1,4}(D)$ | $\mathcal{T}_{64,1,1,1,1,1,1,64}^{2,1,1,1,1,1,1,4}(D)$ | $\mathcal{T}_{64,1,1,1,1,1,1,64}^{2,1,1,1,1,1,1,4}(D)$ |
| (11008,8) | $\mathcal{T}_{2,1,1,1,1,1,1,4}^{86,1,1,1,1,1,1,128}(D)$ | $\mathcal{T}_{2,1,1,1,1,1,1,4}^{86,1,1,1,1,1,1,128}(D)$ | $\mathcal{T}_{2,1,1,1,1,1,1,4}^{86,1,1,1,1,1,1,128}(D)$ |
| (8,11008) | $\mathcal{T}_{86,1,1,1,1,1,1,128}^{2,1,1,1,1,1,1,4}(D)$ | $\mathcal{T}_{86,1,1,1,1,1,1,128}^{2,1,1,1,1,1,1,4}(D)$ | $\mathcal{T}_{86,1,1,1,1,1,1,128}^{2,1,1,1,1,1,1,4}(D)$ |
| **Llama 3.1-8B (Grattafiori et al., 2024)** | | | |
| **LoRa-Over-SVD** | | | |
| (4096,8) | $\mathcal{T}_{2,4}^{64,64}(D)$ | $\mathcal{T}_{2,4}^{64,64}(D)$ | $\mathcal{T}_{2,4}^{64,64}(D)$ |
| (8,4096) | $\mathcal{T}_{64,64}^{2,4}(D)$ | $\mathcal{T}_{64,64}^{2,4}(D)$ | $\mathcal{T}_{64,64}^{2,4}(D)$ |
| (1024,8) | $\mathcal{T}_{2,4}^{32,32}(D)$ | $\mathcal{T}_{2,4}^{32,32}(D)$ | $\mathcal{T}_{2,4}^{32,32}(D)$ |
| (8,1024) | $\mathcal{T}_{32,32}^{2,4}(D)$ | $\mathcal{T}_{32,32}^{2,4}(D)$ | $\mathcal{T}_{32,32}^{2,4}(D)$ |
| (14336,8) | $\mathcal{T}_{2,4}^{112,128}(D)$ | $\mathcal{T}_{2,4}^{112,128}(D)$ | $\mathcal{T}_{2,4}^{112,128}(D)$ |
| (8,14336) | $\mathcal{T}_{112,128}^{2,4}(D)$ | $\mathcal{T}_{112,128}^{2,4}(D)$ | $\mathcal{T}_{112,128}^{2,4}(D)$ |
| **LoRa-Over-MPO** | | | |
| (4096,8) | $\mathcal{T}_{2,1,1,1,1,1,1,4}^{64,1,1,1,1,1,1,64}(D)$ | $\mathcal{T}_{2,1,1,1,1,1,1,4}^{64,1,1,1,1,1,1,64}(D)$ | $\mathcal{T}_{2,1,1,1,1,1,1,4}^{64,1,1,1,1,1,1,64}(D)$ |
| (8,4096) | $\mathcal{T}_{64,1,1,1,1,1,1,64}^{2,1,1,1,1,1,1,4}(D)$ | $\mathcal{T}_{64,1,1,1,1,1,1,64}^{2,1,1,1,1,1,1,4}(D)$ | $\mathcal{T}_{64,1,1,1,1,1,1,64}^{2,1,1,1,1,1,1,4}(D)$ |
| (1024,8) | $\mathcal{T}_{2,1,1,1,1,1,1,4}^{32,1,1,1,1,1,1,32}(D)$ | $\mathcal{T}_{2,1,1,1,1,1,1,4}^{32,1,1,1,1,1,1,32}(D)$ | $\mathcal{T}_{2,1,1,1,1,1,1,4}^{32,1,1,1,1,1,1,32}(D)$ |
| (8,1024) | $\mathcal{T}_{32,1,1,1,1,1,1,32}^{2,1,1,1,1,1,1,4}(D)$ | $\mathcal{T}_{32,1,1,1,1,1,1,32}^{2,1,1,1,1,1,1,4}(D)$ | $\mathcal{T}_{32,1,1,1,1,1,1,32}^{2,1,1,1,1,1,1,4}(D)$ |
| (14336,8) | $\mathcal{T}_{2,1,1,1,1,1,1,4}^{112,1,1,1,1,1,1,128}(D)$ | $\mathcal{T}_{2,1,1,1,1,1,1,4}^{112,1,1,1,1,1,1,128}(D)$ | $\mathcal{T}_{2,1,1,1,1,1,1,4}^{112,1,1,1,1,1,1,128}(D)$ |
| (8,14336) | $\mathcal{T}_{112,1,1,1,1,1,1,128}^{2,1,1,1,1,1,1,4}(D)$ | $\mathcal{T}_{112,1,1,1,1,1,1,128}^{2,1,1,1,1,1,1,4}(D)$ | $\mathcal{T}_{112,1,1,1,1,1,1,128}^{2,1,1,1,1,1,1,4}(D)$ |

Table S.4: Summary of the MPO structure (Llama 2-7B and Llama 3.1-8B).

| Learning Rate | 1.6e-4 | 1.8e-4 | 2e-4 | 2.2e-4 | 2.4e-4 |
|---|---|---|---|---|---|
| **CoLA** | 78.43 | 78.62 | 79.19 | 79.29 | 79.10 |
| **MRPC** | 77.21 | 78.43 | 77.70 | 77.94 | 78.68 |

Table S.5: Comparison of different learning rates on CoLA and MRPC tasks using LoRa-Over-MPO on T5-Base (in percent).

## C.5 LIMITATIONS

The robustness of this over-parameterization framework warrants systematic investigation in future research. Our findings are inherently limited to the NLP tasks and datasets selected for evaluation. Despite adopting established classifications, the selection of downstream tasks and datasets remains inherently subjective, and we do not assess our method on other datasets. Therefore, our work might not be applicable across all benchmarks. Furthermore, due to computational resource constraints, we have not assessed LoRa-Over on larger pre-trained models.