# OpenReview forum: "LoRa-Over: A Matrix Decomposition-Based Over-Parameterization for Efficient LLM Fine-Tuning"
_ICLR.cc/2026/Conference — ICLR 2026 Conference Withdrawn Submission_

### Official Review · Reviewer_cL9B · 2025-10-28

**Soundness:** 2
**Presentation:** 2
**Contribution:** 3
**Rating:** 4
**Confidence:** 4

**Summary:**

This paper proposes a novel over-parameterization method named LoRa-Over, which enhances the generalization performance of Low-Rank Adaptation (LoRA) via Matrix Product Operator (MPO) decomposition. The method demonstrates superior performance over the original LoRA across multiple natural language processing tasks. However, the paper exhibits significant weaknesses in several key areas. The arguments for its novelty are insufficient, the experimental evaluation lacks thoroughness, methodological details are unclear, and the writing lacks rigor. Particularly, the absence of efficiency metrics and inadequate comparison with state-of-the-art baselines substantially undermine the reliability of the conclusions.

**Strengths:**

1. This paper integrates MPO decomposition with over-parameterization for LoRA fine-tuning, proposing static and dynamic selection strategies that significantly enhance the representational capacity of low-rank matrices. The flexible framework offers a novel approach to parameter-efficient fine-tuning.
2. The method is validated on multiple mainstream tasks and models, with results demonstrating that LoRa-Over outperforms the original LoRA on most tasks, particularly showing significant improvements on GSM8K and HumanEval.

**Weaknesses:**

1. The application of MPO decomposition is already established in areas like model compression and Mixture-of-Experts (MoE) frameworks. The paper fails to adequately clarify the fundamental distinctions between LoRa-Over and these similar existing works. The description of the core innovation remains overly vague, leading to a potential overstatement of the paper's contribution. Furthermore, while the static and dynamic selection strategies draw inspiration from model pruning techniques, the paper does not convincingly argue their unique advantages compared to other adaptive methods.
2. A central claim of the paper is "efficient fine-tuning." However, it provides no empirical data on efficiency, such as inference speed (e.g., tokens per second), GPU memory footprint, or FLOPs comparison. ​​This constitutes a major flaw​​, as the absence of these metrics fundamentally undermines the support for the "efficiency" claim and makes it impossible to evaluate the practical utility of the proposed method.
3. The experimental validation is limited to models of up to 8 billion parameters. The scalability of the method remains unverified on larger-scale models (e.g., 32B parameters or more), which is crucial for assessing its broader applicability and impact in the field of large language model fine-tuning.
4. The dynamic selection strategy necessitates the continuous computation of gradient importance scores, which likely introduces additional training time overhead. The paper lacks a quantitative analysis of this computational cost. Additionally, while the dynamic selection is based on a first-order Taylor expansion approximation for scoring, the paper does not validate the accuracy or reliability of this "routing" mechanism for selecting the most critical matrices.
5. The definition and calculation process of the " significance score" depicted in Figure 1 are ambiguous. The main text does not provide a detailed, step-by-step explanation of how this crucial metric is computed, which hinders reproducibility and a clear understanding of the selection mechanism.

**Questions:**

Please refer to the Weaknesses.

---

### Official Review · Reviewer_M4uT · 2025-10-31

**Soundness:** 2
**Presentation:** 2
**Contribution:** 2
**Rating:** 4
**Confidence:** 3

**Summary:**

This paper introduces LoRa-Over, a method that enhances Low-Rank Adaptation (LoRA) by strategically over-parameterizing low-rank matrices during fine-tuning using Matrix Product Operator (MPO) decomposition from quantum physics. The key idea is to temporarily expand the parameter space during training through tensor decomposition, then contract the tensors back to maintain inference efficiency. The authors propose both static and dynamic strategies for selecting which matrices to over-parameterize and demonstrate improvements over vanilla LoRA on various NLP tasks.

**Strengths:**

1 - Applying MPO decomposition from quantum many-body physics to over-parameterize LoRA matrices appears to be a new contribution to the parameter-efficient fine-tuning literature.

2 - The paper evaluates on diverse tasks (NLU with GLUE, dialogue with MT-Bench, mathematical reasoning with GSM8K, code generation with HumanEval) and multiple models (T5-Base, Llama 2-7B, Llama 3.1-8B).

3 - Providing both static (pre-computation based on loss impact) and dynamic (gradient-based during training) strategies offers flexibility.

**Weaknesses:**

1 - The paper does not compare against simply using higher rank in vanilla LoRA with the same parameter budget. This is essential to validate that MPO decomposition specifically provides benefits beyond just having more parameters.

2 - No formal analysis of convergence properties, approximation bounds, or theoretical justification for why MPO specifically helps over other decomposition methods.

3 - The paper never reports the actual training time overhead of MPO decomposition/recomposition operations, which is crucial for practical adoption.

4 - Results report means over 5 runs but provide no confidence intervals, standard deviations, or significance tests. Some improvements are marginal (e.g., 0.11 point on MT-Bench for Llama 3.1-8B).

5 - Terms like "near-lossless reconstruction" are used without quantification. What is the actual reconstruction error? How does it propagate through layers?

**Questions:**

I would appreciate authors elaborating on the following questions:

1 - What is the wall-clock training time overhead compared to vanilla LoRA? Please provide concrete numbers for different model sizes.

2 - How does LoRa-Over compare to vanilla LoRA with rank chosen to match the total parameter count during training? This is a critical missing baseline.

3 - Can you provide confidence intervals and significance tests? Several improvements appear marginal and may not be statistically significant.

4 - Quantify "near-lossless" - what is the actual reconstruction error in Frobenius norm? How does this error accumulate through transformer layers?

5 - Beyond empirical results, what theoretical properties make MPO superior to SVD or other decompositions for this application?

---

### Official Review · Reviewer_SAeR · 2025-10-31

**Soundness:** 1
**Presentation:** 1
**Contribution:** 1
**Rating:** 2
**Confidence:** 4

**Summary:**

This paper proposes an extension of LoRA where low-rank adapters are over-paremeterized using the MPO tensor network parameterization during training. The method is claimed to achieve near loseless reconstruction while maintaining inference time. Static and dynamic strategies to select which adapters to over-parameterize are proposed. The submission is a "method" paper, in the sense that no theoretical results are provided. The proposed method is described from a relatively high level perspective and validated experimentally.

The experiments compare the proposed approach with several variants of LoRA on fine-tuning tasks on some Glue tasks with T5-Base, LLamma 2-7B and LLama 3.1-8B models.

**Strengths:**

- Parameter efficient fine tuning methods are very important and useful and the idea of over-parameterization is interesting and has not been investigated to its full extent yet in the literature.

**Weaknesses:**

The paper is, in my opinion, not ready for publication.

- The mathematical presentation of the tools used in the method contains many innacuracies and is very confusing.
- Not enough details are given on the method to be able to fully understand it or motivate it. I would not be able to reproduce the method (or even explain it in a lecture) from the explanations given in the main paper and appendix.
- Not enough details are given on the baselines (e.g. SVD is used as a baseline but it is not clear how).

**Questions:**

- How are the bond dimension of the MPO decomposition chosen?

- How is inference efficiency preserved despite the over-parameterization?

- How is the low rank structure of the adapter encoded into the MPO?

- Eq. (1) and its surrounding introduction of tensor product is very confusing and to some extent wrong.
   - What does it mean to be an orthonormal basis of a tensor? The orthonormal basis is of the space, not the tensor.
   - There are no tensor contraction in Eq.1. This equation represents the tensor product of two tensors, not their contraction (which would corresponds to summing over a joint index in both tensors to obtain a tensor whose order is 2 less than the some of the orders of T1 and T2).
   - Eq 1 is mixing i and j indices with i1 and i2.

- Idem for Eq 3 and suroundings:
   - The equation is missing the contractions. As it is written, this equation represents the outer (tensor) product of the core tensors of the MPO. To obtain the resulting tensor, one needs to perform m-1 contractions (for each internal edge in the tensor network).
   - What does $\times$ refer to in Eq 4? I am guessing standard multiplication but it is strange (and imo confusing) to use $i_p \times j_p$  instead of  $i_pj_p$ here.

- Could you include the number of parameters of each model in Table 1 and 2 ?

- It is claimed that inference time is equivalent between Lora and the proposed method. Can you provide experimental evidence and theoretical arguments? What about memory usage?

- Please give more details on the SVD baseline you use int he experiment.

- line 135: Hamiltonian should Hadamard

- line 151: missing ". " before "Formally"

- line 161: "by sequentially utilizing the
tensor product operator." -> you need contractions, tensor products are not enough.

- line 258 "Eq. equation"

---

### Official Review · Reviewer_3iza · 2025-11-03

**Soundness:** 1
**Presentation:** 1
**Contribution:** 1
**Rating:** 0
**Confidence:** 5

**Summary:**

The authors propose LoRa-Over, a parameter-efficient fine-tuning method that enhances generalization via strategic over-parameterization of low-rank matrices. LoRa-Over decomposes matrices to selectively over-parameterize critical components, balancing efficiency and performance. Experiments on GLUE, MTBench, GSM8K, and HumanEval show significant improvements over LoRA, demonstrating its effectiveness.

**Strengths:**

-	The idea of using MPO for matrix decomposition is interesting

**Weaknesses:**

-	Limited novelty with potential plagiarism concerns. The proposed method is nearly identical to [1], except for using LoRA fine-tuning.
-	Outdated baselines in Table 1: Comparisons should include more recent methods like LoRA-GA [2] and LoRA-Pro [3]
-	Poor writing and presentation: Figures are unclear, and all mathematical references incorrectly use "Eq. equation x" format

**Questions:**

See above

---

### Note · Authors · 2025-11-13

I have read and agree with the venue's withdrawal policy on behalf of myself and my co-authors.